# Health Economic Decision Tree Models of Diagnostics for Dummies: A Pictorial Primer

**DOI:** 10.3390/diagnostics10030158

**Published:** 2020-03-14

**Authors:** Tamlyn Rautenberg, Annette Gerritsen, Martin Downes

**Affiliations:** 1Centre for Applied Health Economics, Griffith University, Nathan 4111, Australia; m.downes@griffith.edu.au; 2EpiResult, Consultancy, Pietermaritzburg 3201, South Africa; annette.gerritsen@epiresult.com

**Keywords:** decision analysis, health economic modelling, diagnostic test

## Abstract

Health economics is a discipline of economics applied to health care. One method used in health economics is decision tree modelling, which extrapolates the cost and effectiveness of competing interventions over time. Such decision tree models are the basis of reimbursement decisions in countries using health technology assessment for decision making. In many instances, these competing interventions are diagnostic technologies. Despite a wealth of excellent resources describing the decision analysis of diagnostics, two critical errors persist: not including diagnostic test accuracy in the structure of decision trees and treating sequential diagnostics as independent. These errors have consequences for the accuracy of model results, and thereby impact on decision making. This paper sets out to overcome these errors using color to link fundamental epidemiological calculations to decision tree models in a visually and intuitively appealing pictorial format. The paper is a must-read for modelers developing decision trees in the area of diagnostics for the first time and decision makers reviewing diagnostic reimbursement models.

## 1. Introduction

Decision tree models have been used for decades to evaluate diagnostics for the purpose of reimbursement and decision making. Numerous resources are available describing the nuances of diagnostic decision modelling [1,2,3,4,5,6,7,8,9,10,11]. Despite these, decision models of diagnostics continue to have two important weaknesses. For example, a review of thirty economic evaluations for diagnostics in oncology showed that only twelve evaluations modelled diagnostic test accuracy (DTA); the remaining eighteen models only considered the cost of diagnostics and not DTA [4]. This is important because modelling diagnostics without considering DTA is akin to modelling a drug without considering efficacy. It has been shown that models that (correctly) include DTA have higher incremental cost effectiveness ratios and are therefore less likely to be cost-effective when compared to models that do not include DTA; therefore, not including DTA potentially impacts on reimbursement decisions [2,4]. Some potential reasons for not including DTA are the questionable quality of primary sources of evidence which assess DTA, the methods used to synthesize DTA data, and the lack of consensus on the best practices for the modelling of diagnostics [5]. Another recently published review evaluated diagnostic decision models published between the period 2009 and 2018 [12]. Out of fifty-five studies, only thirty-two percent considered the correlation between sequential diagnostics [12]. Both studies evaluated published models only, so, assuming that most of the models which inform reimbursement are not published, many more models may be prone to similar weaknesses. This could have important repercussions on decision making.

In practice, it is common for health economists not used to modeling diagnostics to be assigned to develop diagnostic models with challenging timelines and insufficient time to fully understand the relevant issues. Just as a lack of perfect evidence does not prevent decision making, lack of experience does not prevent inexperienced health economists from modelling diagnostics. For decision makers, it may be difficult to tease out the major weaknesses in a diagnostics model without many hours of reviewing the literature. 

This paper provides an intuitively appealing explanation of the fundamental concepts of the decision analysis of diagnostics in a format easily accessible to modelers applying their skills to diagnostics for the first time. To the best of our knowledge, no current resource links the epidemiological concepts of the 2 × 2 table and the decision tree framework to increase the intuitive appeal of the modelling concepts. For the purpose of this paper, a diagnostic is any intervention/technology or test which classifies patients as test-positive or test-negative to indicate whether the test suggests that they have a disease or not. 

## 2. Two Important Measures of Diagnostic Test Accuracy for Decision Tree Modelling 

When undertaking health economic modelling of pharmaceuticals, it is mandatory to include efficacy and/or effectiveness estimates to capture the clinical benefit of one drug versus another. Similarly, a means to quantify the “efficacy/effectiveness” of a diagnostic is essential when modelling diagnostics and arguably the simplest way for it to be included in a model is by using DTA. The way that DTA is used in clinical practice for medical decision making and patient management is different to how DTA is used in health economics to inform model development; the focus of this paper is on the latter. In addition, there are different ways to quantify and describe DTA, however, this paper focuses on two measures which are important for decision modelling. The first measure consists of paired sensitivity and specificity data which have been derived according to a gold standard. For the purposes of modelling, sensitivity and specificity are commonly sourced from journal publications. Including sensitivity and specificity in a model gives the count of true positives (TP), false positives (FP), true negatives (TN) and false negatives (FN) as a direct result of the diagnostic test. Sensitivity can most accurately be defined as the “probability of correctly identifying, solely from among people who are known to have a condition, all those who do indeed have that condition (i.e., identifying true positives), and, at the same time, not categorizing other people as not having the condition when in fact they do have it (i.e., avoiding false negatives)”(T+|D+) [13]. Specificity can most accurately be defined as the “probability of correctly identifying, solely from among people who are known not to have a condition, all those who do indeed not have that condition (i.e., identifying true negatives), and, at the same time, not categorizing some people as having the condition when in fact they do not have it (i.e., avoiding false positives)”(T−|D−) [13]. 

The second DTA measure of relevance for this paper is paired positive predictive value (PPV) and negative predictive value (NPV). PPV is a measure of patients who, conditional on testing positive (T+), actually are disease-positive (D+|T+), and NPV is a measure of patients who, conditional on testing negative (T−) actually are disease-negative (D−|T−). Unlike sensitivity and specificity, PPV and NPV are dependent upon prevalence and it is important that the prevalence used corresponds to the PPV and NPV values. The reader should be aware that prevalence reported in secondary sources often refers to the general population which, for example, would have a lower PPV and higher NPV compared to clinical practice, is important. Therefore, some calculations or assumptions may be necessary to estimate the prevalence of the disease which corresponds to the scenario being modelled. This is especially relevant for sequential testing (one diagnostic test followed by a second diagnostic test). It is important to know that measures of DTA relate specifically to the conditions under which they are measured, and assumptions may need to be made when applying them to decision trees in the figures modelling scenarios [14]. 

## 3. Decision Tree Model Structures for Diagnostics

Prior to deciding which model structure to use, it is necessary to determine what role the diagnostic will play in the clinical pathway. One scenario occurs when the new diagnostic replaces the current diagnostic, and this will require that the scenarios are modelled in parallel, namely the current clinical pathway with the current diagnostic (scenario 1) modelled in parallel to (compared to) the current clinical pathway but with the new diagnostic in place of the current diagnostic (scenario 2). Alternately, there are two situations when the diagnostics need to be modelled in sequence, one after the other. This may occur when the new diagnostic will be used before the current diagnostic to determine who will go on to be tested with the current diagnostic (triage patients). Alternately, it may occur when the new diagnostic is to be used as an add-on, after the current diagnostic [15]. This may be the case when the subsequent test is comparatively costly and will therefore only be performed in a subgroup of patients who have already had the first gatekeeper test. In either instance, it is necessary to model the diagnostics in sequence.

This paper focuses on two standard decision analytic approaches to decision modelling diagnostics. The first is referred to as a test-based modelling approach and is process-ordered, which means that the diagnostic test is performed first without prior knowledge of who has the disease or not. According to this model structure, the test is done first, and the disease is revealed afterwards, which is the same as the situation in clinical practice. The second approach is referred to as a disease-based modelling approach and begins with a cohort of patients where the disease status is known, based on a gold standard or reference test [13]. In this model structure, the disease status is based on disease prevalence and is made explicit prior to testing, which may be the case in a randomized trial. Schematics of the disease-based (left) and test-based (right) approaches are shown in Figure 1. 

If the prevalence, sensitivity and specificity are available, then a disease-based modelling approach is possible. If PPV, NPV and corresponding prevalence are available, then a test-based modelling approach is possible. 

## 4. Putting Diagnostic Test Accuracy Data and Decision Tree Structures Together: A Worked Example

Set out below is a pictorial worked example to illustrate how prevalence, sensitivity and specificity can be used to populate a 2 × 2 table, which, in turn, can be used to calculate PPV and NPV and parameterize a model. Color is used to link the data in the 2 × 2 tables to the values in the decision tree models so that the connection between the epidemiological data and the decision tree parameters can be made. These estimates can be used to parameterize the two decision tree structures: test-based and disease-based. 

### 4.1. Identifying DTA Data for the Model

It is common for a decision tree model to be developed based on multiple published (secondary) sources. In our worked example, let us assume that the prevalence, and sensitivity and specificity of a diagnostic are reported in a meta-analysis. For exemplar purposes, only a prevalence of 0.4 (40%) is used to simplify the rounding and results of the calculations in a cohort of 100 patients. The prevalence, sensitivity and specificity used in the examples are presented in Table 1.

PPV and NPV, along with the corresponding prevalence, may be reported, but more commonly they will need to be calculated. With the data in Table 1, a 2 × 2 table can be constructed using a cohort of 100 patients. Refer to Table 2 and Table 3 in parallel, and work through the steps in Table 2 to complete the 2 × 2 table shown in Table 3. 

Table 4 is a means of double-checking the completion of the 2 × 2 table. 

Table 5 shows the final steps to calculate the positive predictive value and negative predictive value and the proportion of patients that T+, which is needed for the model.

If sensitivity and specificity are not reported and only TP, FP, TN and FN are reported, then sensitivity and specificity can be back-calculated from the 2 × 2 table. Working through the steps above means that all the data are on hand to parameterize the decision tree models shown below. 

### 4.2. Parameterizing the Decision Tree Model 

Based on the values calculated in Table 3 and Table 5, either decision tree layout shown in Figure 1 can be populated, as shown in Figure 2. Use the colors in Figure 2 to note how the reported data in Table 1 correspond to the parameters required for the disease-based approach (left image in Figure 2) and how the calculated data in Table 5 correspond to the parameters required for the test-based approach (right image in Figure 2). 

All the necessary transition probabilities are available and can be inserted into the model, as shown in Figure 3. 

Use the colors to note how the reported data in Table 1 correspond to the parameters required for the disease-based approach (left image in Figure 3) and how the calculated data in Table 5 correspond to the parameters required for the test-based approach (right image in Figure 3). 

The decision tree in Figure 3 shows the simplest outcome of a diagnostic model, which is the number of TP, FP, TN and FN. These outcomes are intermediate outcomes; they are the direct result of doing the diagnostic test only. These results, however, are important. Note that two of these intermediate outcomes are favorable (TP, TN) and two are unfavorable (FP, FN), therefore it is necessary to consider this when calculating the results of the model. This may be done by representing the favorable outcomes by the number one (TP, TN) and the unfavorable outcomes by zero (FP, FN). 

### 4.3. Linking the Information in the 2 × 2 Table to the Decision Tree

If we insert the cohort of 100 into the decision tree, we can use the decision tree to calculate the numbers shown in the 2 × 2 table, as shown in Figure 4. Use the colors to note how the cohort numbers in Table 3 correspond to the cohort numbers for the disease-based approach (left image in Figure 4) and test-based approach (right image in Figure 4) in the decision-analytic models. It is critical to note that, if carried out correctly, the disease-based and test-based approaches will give the same model results.

The model structures above are used when the new diagnostic will replace the current diagnostic, which requires that the diagnostics are modelled in parallel; however, as described above, sometimes diagnostics are used in sequence. 

## 5. Sequential Diagnostics (Modelling Diagnostics as Triage or Add-on)

When diagnostics are performed in sequence it is necessary to consider that the results of the second diagnostic will be influenced by using the first diagnostic. Failure to recognize the dependency of diagnostics may lead to incorrect results [16,17]. Diagnostics in sequence may be used in the situation where the new diagnostic will be used before the current diagnostic (triage) or after the current diagnostic (add-on), so these examples are referred to as diagnostic 1 and 2. Note that “diagnostic 1” may represent the current or a new diagnostic. The highest level of evidence would be derived from a study which reports the sensitivity and specificity of the two diagnostics used in sequence, therefore the sensitivity and specificity of the second diagnostic is derived in the clinical situation where the second diagnostic is performed in sequence to the first diagnostic (so it already considers the results of the diagnostic performed first). This type of study and data are seldom available in the published literature and it may be practical to use the reported sensitivity and specificity data for a second diagnostic which has not been measured in sequence. A 2 × 2 table can be used to calculate the model parameters.

For this example, we have the same data for diagnostic 1 as before, which are shown in Table 1 and Table 2 and the data for diagnostic 2 is shown in Table 6. 

Firstly, we need to know whether diagnostic 2 will be performed when the results of diagnostic 1 are negative or positive (T− or T+ patients). Based on knowing that, we can proceed to calculate the 2 × 2 table for sequential diagnostic 2. For the purpose of this example we will assume that the test will be done on patients who T+ in diagnostic 1. The information for diagnostic 2 is calculated in the same way as diagnostic 1, as per Table 3 and Table 5. Refer to Table 7 and Table 8 in parallel to complete the 2 × 2 table for diagnostic 2, conditional on performing diagnostic 1 first. 

Finally, calculate PPV, NPV and the proportion of patients who T+ the same as before, shown in Table 9.

The decision model structure for both disease-based and test-based approaches are shown in Figure 5. 

Note that, again, if carried out correctly the two approaches will provide the same model results. It is important to note that, for sequential diagnostics, test correlation and conditional dependence are relevant, and some assumptions may need to be considered [18]. 

## 6. Contextualizing the Information in This Paper

There are some important considerations to note when contextualizing the examples presented in this paper. The first is that the decision trees illustrated in this paper form one component (arguably the most important) of a model; however, they will most likely not be the finished model. This is because the value of a diagnostic is directly linked to the downstream treatments/interventions which follow it. It is the critical DTA component presented in this paper that primarily determines the proportion of patients in a cohort model who can potentially benefit from downstream interventions, therefore this is the critical component of the model [9]. The DTA outcomes illustrated in these examples are intermediate outcomes (TN, TP, FN, FP), which are the first direct consequences of performing the diagnostic. Intermediate outcomes may be suitable for some clinical situations, but usually they are followed by longer term final outcomes which, for the purpose of brevity, are not shown in this paper. In the simplest example, if a potentially life-threatening disease is diagnosed, then the intermediate diagnostic outcomes (TN, TP, FN, FP) will be followed by post-treatment final outcomes, such as survival, mortality or life years saved (assuming a curative treatment is available). Some diagnostic models focus on these longer-term outcomes and “miss” the intermediate outcomes. However, as mentioned, this is like “missing” the efficacy of a drug when modelling pharmaceuticals. How the intermediate outcomes relate to the final outcomes in a model is important and specific to the clinical and decision context. Good understanding of these is essential but beyond the scope of this paper [14]. The reader is directed to other published models to recognize that the components presented in this paper are embedded in models with a broader context in the case of sequential and parallel diagnostics, respectively [4,5]. 

Secondly, it is worth noting that, generally, no test should claim to have one hundred percent sensitivity and/or specificity, and that even if such claims are made this does not justify excluding the DTA component shown in this paper, which is arguably best practice. By including the functionality of the DTA component, the impact of uncertainty around the claims of sensitivity and specificity can be explored with univariate and multivariate sensitivity analysis across a range of plausible values. There are instances where T+ and T− do not adequately capture the range of possible outcomes of a test. For example, in genetic testing where there are “uncertain” results (variants of unknown clinical significance), this case may warrant the addition of a third outcome: test uncertain (T?) in each decision tree arm, added to the model structures shown in the examples above (beyond the scope of this paper). 

The third consideration is that measures of sensitivity and specificity are based on an underlying cut-off threshold which determines the classification of the results into D+T+, D+T−, D−T+ and D−T−, based on a gold standard test. This threshold can be explicit/quantified or implicit/not quantified and may be referred to as the Optimal Operating Point (OOP) [2,5]. Threshold values can vary according to factors such as laboratory standards, expert consensus, specific settings and sample sizes [8]. The most accurate cost-effectiveness results will be obtained when the sensitivity and specificity corresponding to the OOP is used in a decision model. However, most studies do not report thresholds, therefore the sensitivity and specificity modelled seldom correspond to the OOP. The impact of this has been explored in the literature [4,19,20]. Consider further that if a test threshold for positivity is decreased, the sensitivity of a test will increase and the specificity will decrease (sensitivity and specificity are negatively correlated) [2,21]. This needs to be considered when undertaking multivariate (scenario) analysis, for example. For simplification, the examples in this paper assume an implicit/unquantifiable threshold. As discussed, it is also important to take care when using PPV and NPV. Both measures are calculated according to prevalence and it is essential that the prevalence corresponding to PPV and NPV (which may be reported in a secondary study as general population prevalence) represents the modelled population (possibly by assumption). Therefore, disease-based models have a lower potential for error because they rely on sensitivity and specificity estimates compared to test-based models, which have higher potential for error because they rely on PPV and NPV with linked prevalence data. 

The fourth consideration is that there are other measures of DTA beyond the two discussed in this paper. Three of them require sensitivity and specificity data: likelihood ratios (for positive test result, likelihood ratio equals sensitivity/(1-specificity), for negative, test result likelihood ratio equals (1-sensitivity)/specificity)); diagnostic odds ratios ((TP/FN)/(FP/TN)) and Youden’s index (sensitivity + specificity)-1 [22]. The reader should also be aware of other health economic approaches to comparing diagnostics which are described in the literature, including value of information analysis specifically relating to diagnostics (not to be confused with traditional health economic value of information analysis) and ROTS analysis [3]. Another means of comparing DTA is a receiver operating characteristic (ROC) curve, but THIS requires that the OOP is known. ROC’s graph sensitivity versus one minus specificity. However, when graphing multiple diagnostics, an ROC curve may crossover, making it difficult to determine which diagnostic is optimal. ROC curves do not capture costs, side effects and FP/FN, hence the need for decision models [3,5]. ROC curves are further limited because they may not be possible for diagnostics with intangible thresholds or for individual studies, and hence a summary ROC curve may be required [5]. 

The final miscellaneous considerations are that standard meta-analytical approaches for pooling proportions can be applied to diagnostic sensitivity and specificity, but they assume both are constant and independent [16]. There are nuances to undertaking meta-analysis of diagnostics, and the methods are more complex than for treatments which come from randomized controlled trials [5,23,24,25,26]. The reader is directed to Doble et al.’s published “checklist for including characteristics of companion diagnostics into economic evaluations”, which is a valuable resource [4,16,27,28]. It is worth noting that commonly used economic evaluation quality grading tools may fail to adequately capture weaknesses in diagnostic economic evaluations [4]. 

## 7. Conclusions

The pictorial examples in this paper address two fundamental weaknesses in diagnostic decision trees: not including DTA and not considering the dependence between sequential tests. These two issues have been simplified and made explicit with the aim of improving quality, understanding and, ultimately, decision making around diagnostics.

## Figures and Tables

**Figure 1 diagnostics-10-00158-f001:**
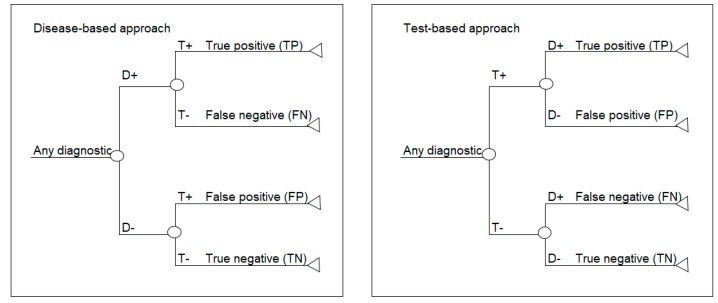
Decision-analytic model structure of disease-based and test-based approaches to modelling diagnostics. D+ = disease positive; D− = disease negative; T+ = test positive; T− = test negative; TP = true positive; FP = false positive; TN = true negative; FN = false negative

**Figure 2 diagnostics-10-00158-f002:**
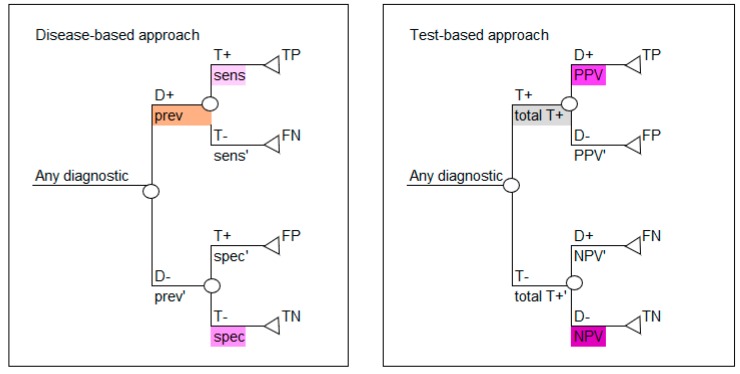
Values derived in Table 3 and Table 5 are used to parameterize the disease-based and test-based decision tree structures. D+ = disease positive; D− = disease negative; T+ = test positive; T− = test negative; sens = sensitivity; sens’ = complement of sensitivity which is 1-sens; spec = specificity; spec’ = complement of specificity which is 1-spec; PPV = positive predictive value; PPV’ = complement of positive predictive value which is 1-PPV; NPV = negative predictive value; NPV’ = complement of negative predictive value which is 1-NPV; all T+ = all test positives; all T+’ = complement of all test positives which is 1-all T+; TP = true positive; FP = false positive; TN = true negative; FN = false negative. The colors in all tables are used to identify types of data and the corresponding data and colors are shown in all figures.

**Figure 3 diagnostics-10-00158-f003:**
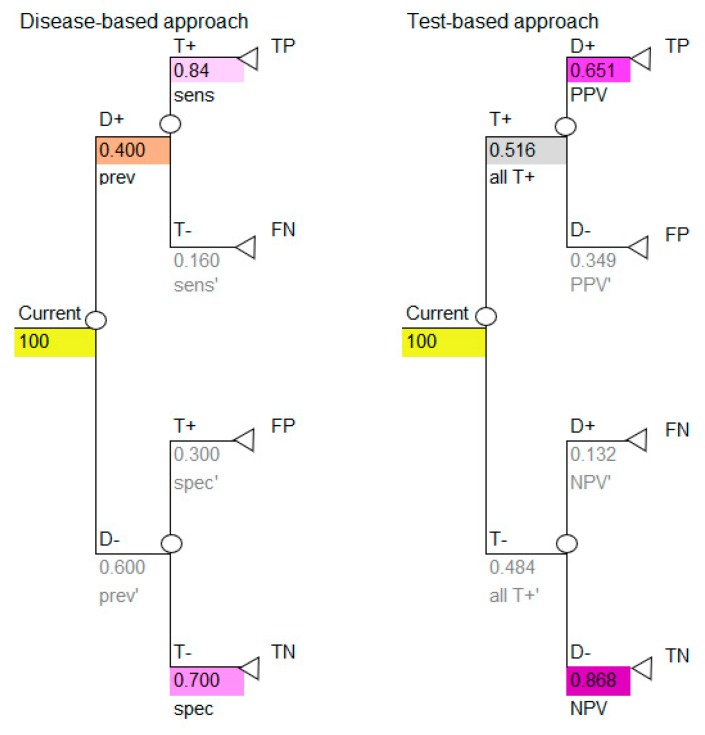
Inserting the probabilities into the model: disease-based and test-based approaches. D+ = disease positive; D− = disease negative; T+ = test positive; T− = test negative; sens=sensitivity; sens’ = complement of sensitivity which is 1-sens; spec = specificity; spec’ = complement of specificity which is 1-spec; PPV = positive predictive value; PPV’ = complement of positive predictive value, which is 1-PPV; NPV = negative predictive value; NPV’ = complement of negative predictive value, which is 1-NPV; all T+ = all test positives; all T+’ = complement of all test positives which is 1-all T+; TP = true positive; FP = false positive; TN = true negative; FN = false negative. The colors in all tables are used to identify types of data and the corresponding data and colors are shown in all figures.

**Figure 4 diagnostics-10-00158-f004:**
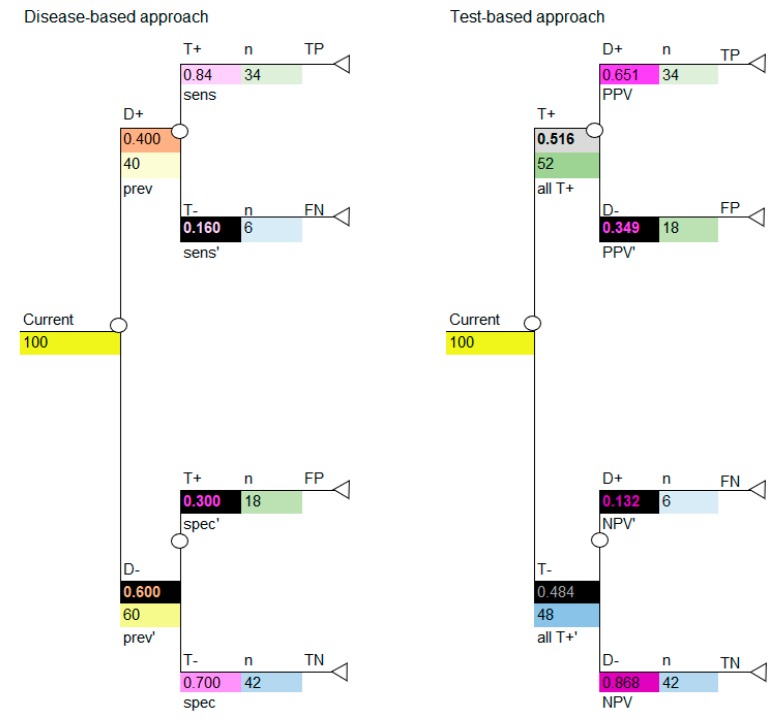
Decision tree, populated with probabilities, calculating number of patients according to a cohort of 100 to correspond to a 2 × 2 table, with outcomes shown for disease-based and test-based approaches. Current versus new diagnostic are modelled in parallel. D+ = disease positive; D− = disease negative; T+ = test positive; T− = test negative; sens = sensitivity; sens’ = complement of sensitivity which is 1-sens; spec=specificity; spec’ = complement of specificity which is 1-spec; PPV = positive predictive value; PPV’ = complement of positive predictive value which is 1-PPV; NPV=negative predictive value; NPV’ = complement of negative predictive value which is 1-NPV; all T+ = all test positives; all T+’ = complement of all test positives which is 1-all T+; TP = true positive; FP = false positive; TN = true negative; FN = false negative, n = number (cohort). The colors in all tables are used to identify types of data and the corresponding data and colors are shown in all figures.

**Figure 5 diagnostics-10-00158-f005:**
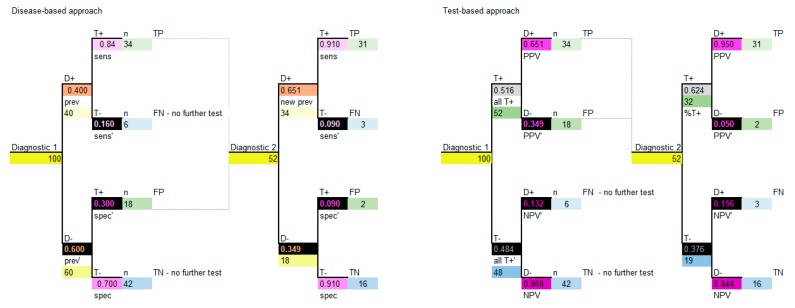
Decision analysis, disease-based and test-based approaches for sequential diagnostics. D+ = disease positive; D− = disease negative; T+ = test positive; T− = test negative; sens = sensitivity; sens’ = complement of sensitivity which is 1-sens; spec = specificity; spec’ = complement of specificity which is 1-spec; PPV = positive predictive value; PPV’ = complement of positive predictive value which is 1-PPV; NPV=negative predictive value; NPV’ = complement of negative predictive value which is 1-NPV; all T+ = all test positives; all T+’ = complement of all test positives which is 1-all T+; TP = true positive; FP = false positive; TN = true negative; FN = false negative, n = number (cohort). The colors in all tables are used to identify types of data and the corresponding data and colors are shown in all figures.

**Table 1 diagnostics-10-00158-t001:** Diagnostic 1 Prevalence, sensitivity and specificity.

Data	Value	Comment
Prevalence	0.400	reported as 40%
Sensitivity	0.840	reported as 84%
Specificity	0.700	reported as 70%

The colors in all tables are used to identify types of data and the corresponding data and colors are shown in all figures.

**Table 2 diagnostics-10-00158-t002:** Steps for calculating the 2 × 2 table using prevalence, sensitivity and specificity.

Step	Objective	Instruction	Cell Notation	Formula
^1^	Define cohort	Insert a cohort of 100 ^a^	(A+B) + (C+D)	na
^2^	Find the total D+	Multiply the prevalence by the cohort to find the number of D+ patients	(A+C)	=0.400 × 100 = 40
^3^	Find the total D−	Subtract the D+ from the total cohort to find D−	(B+D)	=100−40 = 60
^4^	Find D+T+ (TP)	Multiply the sensitivity of the test to the total D+ patients to get the D+T+	A	=0.840 × 40 = 34
^5^	Find D+T− (FN)	Subtract the D+T+ from the total D+	C	=40−34 = 6
^6^	Find D−T− (TN)	Multiply the specificity of the test to the total D− patients to get the D−T−	D	=0.700 × 60 = 42
^7^	Find D−T+ (FP)	Subtract the D−T− from the total D-	B	=60−42 = 18
^8^	Find total T+	Add D+T+ and D−T+	(A+B)	=34 + 18 = 52
^9^	Find total T−	Add D+T− and D−T−	(C+D)	=6 + 42 = 48

^a^ A cohort of one hundred has been used so that the numbers in the 2 × 2 table and the decision trees are easily recognized. D+ = disease positive; D− = disease negative; T+ = test positive; T− = test negative; TP = true positive; FP = false positive; TN = true negative; FN = false negative; na = not applicable. Red letters are cell notation; ^1^^–9^ refer to the superscript numbers in Table 3. The colors in all tables are used to identify types of data and the corresponding data and colors are shown in all figures.

**Table 3 diagnostics-10-00158-t003:** Using prevalence, sensitivity and specificity to complete a 2 × 2 table.

	D+	D−	Total
T+	34 ^4^ (A) TP	18 ^7^ (B) FP	52 ^8^ (A+B) total T+
T−	6 ^5^ (C) FN	42 ^6^ (D) TN	48 ^9^ (C+D) total T−
Total	40 ^2^ (A+C) total D+	60 ^3^ (B+D) total D−	100 ^1^ (A+B) + (C+D) cohort

D+ = disease positive; D− = disease negative; T+ = test positive; T− = test negative; TP = true positive; FP = false positive; TN = true negative; FN = false negative; red letters are cell notation (how the cells are named); superscript numbers indicate the order in which the table is completed. The colors in all tables are used to identify types of data and the corresponding data and colors are shown in all figures.

**Table 4 diagnostics-10-00158-t004:** Steps to double-check completion of the 2 × 2 table.

Step	Objective	Instruction	Formula
10	Check the sensitivity using the 2 × 2 table (checking your work)	Divide A by (A+C)	=34/40 = 0.840
11	Check the specificity using the 2 × 2 table (checking your work)	Divide D by (B+D)	=42/60 = 0.700

Red letters are cell notation.

**Table 5 diagnostics-10-00158-t005:** Using the 2 × 2 table to calculate positive predictive value (PPV), negative predictive value (NPV) and proportion of cohort who T− and T+, respectively.

Step	Objective	Cell	Instruction	Formula
12	Calculate PPV	N/A	Divide A by (A+B)	=34/52 = 0.651
13	Calculate NPV	N/A	Divide D by (C+D)	=42/48 = 0.868
14	Calculate proportion of cohort who T+	N/A	Divide total T+ by cohort	=52/100 = 0.516

Red letters are cell notation. The colors in all tables are used to identify types of data and the corresponding data and colors are shown in all figures.

**Table 6 diagnostics-10-00158-t006:** Diagnostic 2: Sensitivity and specificity.

Data	Value	Source	Comment
Sensitivity	0.910	Meta-analysis	reported as 91%
Specificity	0.910	Meta-analysis	reported as 91%

The colors in all tables are used to identify types of data and the corresponding data and colors are shown in all figures.

**Table 7 diagnostics-10-00158-t007:** Calculation of 2 × 2 table for diagnostic 2, based on diagnostic 1 being performed first, for sequential diagnostics.

Step	Instruction	Value		Value
1	Start by taking the data from the diagnostic 1 T+ and inserting them into the test result row for diagnostic 2			
2	So the D+T+ for diagnostic 1	34	becomes the D+ for diagnostic 2	34
	The D−T+ for diagnostic 1	18	becomes the D− for diagnostic 2	18
3	The total T+ for diagnostic 1	52	becomes the cohort for diagnostic 2	52
	Based on total D+ (A+C) and the cohort (A+B) + (C+D), calculate new prevalence = 34/52 = 0.651			
	New prevalence			0.651 ^1^

^1^ Note that the prevalence is now higher than before, because we have “filtered out” those with disease when using diagnostic 1 (more of this new cohort D+ than before). Note that the sample size is now 52 and the new prevalence is based on this new sample size (34/52 = 0.651, which is 65.1%). The colors in all tables are used to identify types of data and the corresponding data and colors are shown in all figures.

**Table 8 diagnostics-10-00158-t008:** Calculating the 2 × 2 table for diagnostic 2 when done in sequence to diagnostic 1.

	D+	D−	Total
T+	31 ^4^ (A) TP	2 ^7^ (B) FP	32 ^8^ (A+B) total T+
T-	3 ^5^ (C) FN	16 ^6^ (D) TN	19 ^9^ (C+D) total T−
Total	34 ^1^ (A+C) total D+	18 ^2^ (B+D) total D-	52 ^3^ (A+B) + (C+D) cohort

D+ = disease positive; D− = disease negative; T+ = test positive; T− = test negative; TP = true positive; FP = false positive; TN = true negative; FN = false negative; Red letters are cell notation, how the cells are named. Superscript numbers indicate the order in which the table is completed. The colors in all tables are used to identify types of data and the corresponding data and colors are shown in all figures.

**Table 9 diagnostics-10-00158-t009:** Using the 2 × 2 table to calculate PPV, NPV and the proportion of cohort who T− and T+, respectively, for diagnostic 2.

Step	Objective	Cell	Instruction
12	Calculate PPV	N/A	Divide A by (A+B)
13	Calculate NPV	N/A	Divide D by (C+D)
14	Calculate proportion of cohort who T+	N/A	Divide total T+ by cohort
15	Calculate proportion of cohort who T−	N/A	Divide total T- by cohort

T+ = test positive; T− = test negative; PPV = positive predictive value; NPV = negative predictive value. Red letters are cell notation. The colors in all tables are used to identify types of data and the corresponding data and colors are shown in all figures.

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
