# Peer review of "Health Economic Decision Tree Models of Diagnostics for Dummies: A Pictorial Primer"

_diagnostics, 2020, doi:10.3390/diagnostics10030158_

Round 1
Reviewer 1 Report
The paper is well-written, scientifically sound and extremely helpful. I commend its acceptance.
Author Response
We thank you for your favorable feedback.
Reviewer 2 Report
Dear authors.
Be congratulated on this timely and very pedagogical contribution! The paper discusses the main challenge of employment of diagnostic accuracy figures in health modelling excellently and contrasts differences to health economic modelling of drugs (e.g. accuracy as an intermediate efficacy outcome and its necessary translation into patient-related outcomes like survival of progression-free survival). Limitations of the study are transparently delineated for the reader.
I attached the review PDF in which I added some minor comments and (literature) proposals. Please check numbers for TN in Fig. 5 which deviates in left and right part of the figure. Moreover, please revisit the formatting of the reference list which needs correcting (see comments and, for instance, #11: “Hunink, M.G.G.P.S.J.W.J.P.J.E.A.W.M., Decision…”). Generally, I wondered why large parts of the text were highlighted with yellow background, for instance, nearly the whole abstract.

Author Response
We thank you for your favorable feedback.
We have directly changed the text in response to your comments (punctuation, adding a reference and reformatting the reference list) except for the following two instances:
- Reviewer 2’s pdf, line 271 – the comment: “in this paper (think, for instance, of survival or time-to-progression in cancer trials)”.
- Author response: We were reluctant to add the final outcomes for cancer because although it is easy to link them to a treatment it is more difficult to link them directly to a diagnostic test. Therefore to address this comment we added a simplified example: “In the simplest example, if a potentially life-threatening disease is diagnosed (assuming a curative treatment is available) then the intermediate diagnostic outcomes (TN, TP, FN, FP) will be followed by post-treatment final outcomes such as survival, mortality or life years saved” shown in line 273-288 of the track changes and clean version.
- Reviewer 2’s pdf, line 284 the comment: “with sensitivity analysis (e.g. by employing lower bounds of 95% confidence intervals for, say, sensitivity and specificity)”.
- Author response: this comment highlighted to us that this sentence in the paper was unclear so it has been completely revised as follows: “Secondly, it is worth noting that generally no test should claim to have one hundred percent sensitivity and/or specificity, and that even if such claims are made this does not justify excluding the DTA component shown in this paper, which is arguably best practice. By including the functionality of the DTA component, the impact of uncertainty around the claims of sensitivity and specificity can be explored with univariate and multivariate sensitivity analysis across a range of plausible values.” shown in line 283-288 of track changes and clean versions.
This manuscript is a resubmission of an earlier submission. The following is a list of the peer review reports and author responses from that submission.
Round 1
Reviewer 1 Report
This is a long-awaited paper that explains in a simple, plain language a topic of paramount mportance within the field of evidence-based biomedicine, with practical examples and applications. Readers can follow step by step the different procedures and can learn how to utilize an important tool. Authors should be congratulated for their efforts.
Author Response
We thank the reviewer for their insightful feedback which reflects a sound understanding of the current literature and the issues that the paper addresses.
Reviewer 2 Report
In this study, Rautenberg et al. seek a new concept of decision tree models for diagnostic testing. Diagnostic accuracy is determined by various factors, not only by sensitivity/specificity or PPV/NPV. Many studies represent odds ratio and ROC curve analyses, but the concept introduced in this study completely ignores these analyses. In addition, studies in reality are not as simple as examples shown in this study. Diagnostic accuracy can differ depending on cohorts, races, or institutions. Different studies may use different criteria or procedures for diagnosis or testing. It is impossible to mix all studies and calculate new numbers regardless of those other factors. It is unclear which study should be recognized as “A” or “B” and used in calculation. Doctors and researchers check various studies from various institutes and make a decision. That is reasonable and no techniques can represent 100% sensitivity/specificity. They know one study or number is not enough and results can differ in the different situation because various factors are involved and the disorder is not that simple. The concept the authors represent in this study does not improve anything. The authors claim that this is simple, but this is because they ignore other factors or analyses. For me, this study looks like that the authors merely play with numbers in their brains on the desk and make their story ignoring other things that do not support their story. If the authors claim that this concept is useful, they need to prove it using real studies.
Author Response
|
Reviewer 2 Comment |
Author response |
Manuscript changes |
|
In this study, Rautenberg et al. seek a new concept of decision tree models for diagnostic testing. |
Sorry for any misunderstanding, this is not a new concept but a concept that has been used for decades but used incorrectly. |
The manuscript wording has been improved to make this clearer – please see track changes. |
|
Diagnostic accuracy is determined by various factors, not only by sensitivity/specificity or PPV/NPV. Many studies represent odds ratio and ROC curve analyses, but the concept introduced in this study completely ignores these analyses. |
Our study focuses only on two of the fundamental errors which rely on sens/spec and PPV/NPV. We do contextualise ROC curves. We see the discussion of OR as more complex to understand for beginners therefore we have not discussed it here. |
|
|
In addition, studies in reality are not as simple as examples shown in this study. Diagnostic accuracy can differ depending on cohorts, races, or institutions. Different studies may use different criteria or procedures for diagnosis or testing. It is impossible to mix all studies and calculate new numbers regardless of those other factors. |
We acknowledge that, however we needed to use these very simplified examples to illustrate the core concepts which are often a source of errors. |
|
|
Doctors and researchers check various studies from various institutes and make a decision. That is reasonable and no techniques can represent 100% sensitivity/specificity. They know one study or number is not enough and results can differ in the different situation because various factors are involved and the disorder is not that simple. |
We acknowledge that these decisions are made as such in clinical practice. Health economists use other methods to evaluate diagnostics and decision trees is one of those methods. |
|
|
The concept the authors represent in this study does not improve anything. The authors claim that this is simple, but this is because they ignore other factors or analyses. For me, this study looks like that the authors merely play with numbers in their brains on the desk and make their story ignoring other things that do not support their story. If the authors claim that this concept is useful, they need to prove it using real studies. |
We are sorry that the value of this paper has not been accurately communicated in the original submission and we have modified this revised version accordingly to strengthen this message. |
Please see track changes |
Reviewer 3 Report
The revised manuscript analyzes Models of diagnostic decision tree. The manuscript requires the authors to delve into various sections.
First, the authors have to elaborate the abstract better and indicate the keywords.
Secondly, the authors have to increase the number of bibliographic references of the introduction (at most of the last 5 years) to justify the paper.
Thirdly, it is necessary to improve section 5 and 7, as well as describe more the conclusions of the work.
Finally, on line 145 and 246, there is an error and the bibliography is not shown.
The work requires a major revision in order to deepen the gap of the manuscript.
Author Response
Reviewer 3
|
Reviewer 3 Comment |
Author response |
Manuscript changes |
|
The revised manuscript analyzes Models of diagnostic decision tree. |
Apologies for any misunderstanding, the version reviewed was a first submission, not a “revised manuscript”. |
|
|
The manuscript requires the authors to delve into various sections. First, the authors have to elaborate the abstract better and indicate the keywords. |
|
Abstract expanded, keywords added. |
|
Secondly, the authors have to increase the number of bibliographic references of the introduction (at most of the last 5 years) to justify the paper. |
|
In the bibliography, references number one to nine describe publications that deal with the nuances of modelling of diagnostics. Bibliography items number 10 (Yang 2019), 11 (Trevethan 2017), 12 (Bossuyt 2013), 13 (Ranganathan 2018) and 14 (Macey 2015) are the more recent studies. Unfortunately, there is an absence of literature in this area and no further references are available hence the need for this paper. |
|
Thirdly, it is necessary to improve section 5 and 7, as well as describe more the conclusions of the work. |
|
Conclusion revised – please see track changes. |
|
Finally, on line 145 and 246, there is an error and the bibliography is not shown. |
Sorry I cannot identify the error in line 145. The following text appears in the submitted version line 145 sentence starts in row 144):
The following text appears in line 145 of the online version of the manuscript (version downloaded for revisions from the online link). “Schematics of the disease-based (left) and test-based (right) approaches are shown in Figure 1.” Sorry I cannot identify the error in line 246. The following text appears in the submitted version line 246 Caption for Table 8 “Table 8: Calculating the 2x2 table for diagnostic 2 when done in sequence to diagnostic 1”.
The following text appears in line 246 of the online version of the manuscript (version downloaded for revisions from the online link). “The decision model structure for both disease-based and test-based approaches are shown in Figure 5” In both manuscript versions the bibliography is shown. |
|
|
The work requires a major revision in order to deepen the gap of the manuscript. |
|
Please see track changes as above |
Round 2
Reviewer 2 Report
The authors did not address my comments/concerns. In conclusion, they say "... will go some way to improving the quality of decision ...", but they do not show any evidence. No evidence for improving the quality. This is not science. This is just the authors' imagination. I cannot accept this manuscript.
Reviewer 3 Report
The revised manuscript analyzes Models of diagnostic decision tree. The authors have not improved the quality of the manuscript since the changes I suggested have not been considered in depth (introduction, sections 5 and 7) and only small changes in abstract, keywords and conclusions have been done.
The authors have not changed to the error described (Error! Reference source not found) on lines 153 and 254 of the revised version that also appeared in the first revision (lines 145 and 246).
My recommendation is reject the manuscript.